# Current Insights into Atopic March

**DOI:** 10.3390/children8111067

**Published:** 2021-11-19

**Authors:** Mitsuru Tsuge, Masanori Ikeda, Naomi Matsumoto, Takashi Yorifuji, Hirokazu Tsukahara

**Affiliations:** 1Department of Pediatric Acute Diseases, Okayama University Academic Field of Medicine, Dentistry, and Pharmaceutical Sciences, Okayama 700-8558, Japan; tsugemitsuru@okayama-u.ac.jp; 2Department of Pediatrics, Okayama University Academic Field of Medicine, Dentistry, and Pharmaceutical Sciences, Okayama 700-8558, Japan; mikeda@okayama-u.ac.jp; 3Department of Epidemiology, Okayama University Academic Field of Medicine, Dentistry, and Pharmaceutical Sciences, Okayama 700-8558, Japan; naomim@okayama-u.ac.jp (N.M.); yorichan@md.okayama-u.ac.jp (T.Y.)

**Keywords:** atopic march, atopic dermatitis, food allergy, allergic asthma, allergic rhinitis, skin barrier dysfunction, alarmin, group 2 innate lymphoid cells, type 2 inflammation, oxidative stress, specific biomarker, epidemiology, phenotype, early intervention, emollient

## Abstract

The incidence of allergic diseases is increasing, and research on their epidemiology, pathophysiology, and the prevention of onset is urgently needed. The onset of allergic disease begins in infancy with atopic dermatitis and food allergy and develops into allergic asthma and allergic rhinitis in childhood; the process is defined as “atopic march”. Atopic march is caused by multiple immunological pathways, including allergen exposure, environmental pollutants, skin barrier dysfunction, type 2 inflammation, and oxidative stress, which promote the progression of atopic march. Using recent evidence, herein, we explain the involvement of allergic inflammatory conditions and oxidative stress in the process of atopic march, its epidemiology, and methods for prevention of onset.

## 1. Introduction

### 1.1. Atopic March and Atopic Dermatitis (AD)

“Atopic march” refers to the natural history of allergic diseases. Atopic diseases of different organs and causal allergens develop sequentially with age; some symptoms become more prominent over time, while others subside. AD generally develops first, followed by immunoglobulin E (IgE)-mediated food allergy (FA), allergic asthma (AA), and allergic rhinitis (AR), which result in increased sensitization to food and/or environmental allergens (Figure 1 and Figure 2). AD is regarded as the starting point of atopic march [1,2], and the risks of FA, AA, and AR increase as the severity of AD increases [3,4,5]. AD develops in 17–24% of children, and 48% and 89% of the patients develop AD by 6 months and 5 years of age, respectively. The prevalence of symptoms peaks by two years of age [6]. Furthermore, about half of children with AD go into remission by adolescence, while AD persists into adulthood in the remaining half [1].

### 1.2. AD, FA, and Epicutaneous Sensitization

The risk of FA onset is six times higher in children with AD than in those without AD [5]. The prevalence of FA peaks at about 10% at 1 year of age, and specific IgEs for food allergens are detectable in infants at 1 month of age. In most cases, allergic sensitization appears to occur before oral intake of food. The Learning Early About Peanut Allergy (LEAP) screening study showed that as AD severity (assessed by Scoring Atopic Dermatitis (SCORAD) score) increases, the corresponding rate of cutaneous sensitization to peanuts, hen eggs, and sesame also increases in 4–10-month-old infants [7]. Moreover, 50% of children with early onset and severe AD had challenge-proven FA by the age of 12 months [8]. The risk of FA increases remarkably in early-onset and severe AD, especially within the first 1–4 months after birth [8,9,10].

In addition, positive associations have been shown between the use of skincare products containing wheat or peanuts and the onset of wheat and peanut allergies, respectively [3]. The expression of skin-related homing molecules on allergen-specific T cells isolated from patients with peanut allergy has also been demonstrated [11,12]. Additionally, it has been shown that early aggressive use of topical steroids to shorten the duration of eczema in infants was associated with decreased development of FA [10].

These findings support the presence of epicutaneous sensitization to food allergens in highly sensitive individuals. The results also suggest that early suppression of inflammation and reduced duration of eczema lowers the chances of epicutaneous exposure to environmental food allergens, thereby preventing the sensitization and subsequent development of FA.

### 1.3. AA and AR in Atopic March

Infants and young children with AD are at great risk of developing AA and AR later in life, and this risk ranges from 50 to 75% [13]. The prevalence of AA in children with mild AD is about 20%, but it increases to >60% in those with severe AD [9]. Early onset of AD was also associated with an increased risk of sensitization to inhalant allergens [13] and persistence of AA into adulthood. The risk of AA and AR was associated with age at AD onset and its severity. Unlike sensitization to food allergens, sensitization to inhalant allergens mostly occurs after infancy. Its annual incidence and prevalence increase markedly with age after 3 years of age [14], providing a possible explanation for the late development of AA and AR in atopic march.

Among children with AA, 74–81% have AR [15], and these children tend to manifest AR earlier in life (2.9 ± 1.7 years of age) [16]. AA was reportedly more severe and harder to control in children with both AA and AR, and AA was better controlled when AR was treated. Appropriate management of AR not only alleviates the symptoms of AA, but also improves airway hyperresponsiveness [17]. According to these findings, AA and AR are considered to be “one airway, one disease” [18]. The Guidelines on AR and its Impact on Asthma (ARIA) indicate that proactive intervention for AR is essential for adequately controlling AA [18,19].

In addition, otitis media, which is highly associated with AR, is an additional part of atopic march [20]. Mast cells and eosinophils can cause allergic reactions in the mucosa of the middle ear as well as in that of the airways [21,22,23]. Allergic asthma and rhinitis activate eosinophil cationic protein and other allergy-related mediators in the middle ear [22], and then increase the risk of otitis media with effusion [24].

### 1.4. FA and Atopic March

In children with FA, the risk of developing AA and AR is 2.1–5.3-fold and 1.6–5.1-fold higher than in children without FA, respectively. The risk of developing AA or AR has been reportedly higher in children with multiple FA than in those with a single FA, and was also higher in children with severe FA (e.g., anaphylaxis) than in those with non-severe FA [25]. Sensitization to food allergens before 2 years of age increased the risk of wheeze/AA (pooled odds ratio (OR) 2.9, 95% confidence interval (CI) 2.0–4.0), AD (pooled OR 2.7, 95% CI 1.7–4.4), and AR (pooled OR 3.1, 95% CI 1.9–4.9) [26]. These findings suggest that there is a risk that additional allergic conditions will emerge from existing allergic conditions with the progression of atopic march. Therefore, preventing the onset of atopic march and its progression at an early stage is crucial. There is an urgent need for progress in research aimed at developing more effective early intervention strategies to ameliorate these sequential atopic disorders.

## 2. The Immune Response behind Atopic March

### 2.1. Skin Barrier Dysfunction and Its Role in AD

The pathological mechanism of AD comprises three aspects: skin barrier dysfunction, T helper 2 (Th2) immune skewing, and pruritus. In AD, skin barrier dysfunction increases irritability due to external stimuli, and the skin becomes more prone to allergen sensitization and inflammation [27]. Filaggrin is a major protein that fills the cytoplasm of corneocytes. It aggregates keratin fibers, and its degradation products serve as moisturizing factors, indicating that filaggrin plays an important role in maintaining skin barrier function, retaining moisture, and lowering pH [27]. Abnormal barrier function due to filaggrin gene mutations was confirmed in patients with AD or AA, and filaggrin gene mutations have been shown to increase the risk of developing early and severe AD, as well as AA in patients with AD [28]. In addition, the association of the development of atopic march with abnormal skin barrier function (e.g., due to filaggrin gene mutations) and epithelial cell (EC)-derived cytokines, such as thymic stromal lymphopoietin (TSLP), interleukin (IL)-33, and IL-25, has been suggested [2,29].

### 2.2. The Immune Response of Sensitization in AD and Atopic March

Exposure of skin with impaired barrier function to food allergens and dust mites leads to the release of EC-derived alarmins (TSLP, IL-25, and IL-33) that induce the activation of immature dendritic cells (DCs) and group 2 innate lymphoid cells (ILC2s), thereby triggering innate immune responses [30]. DCs that capture allergens migrate to draining lymph nodes and then process and present allergens to naïve T cells [31]. The naïve T cells differentiate into allergen-specific Th2 cells to produce high levels of IL-4 and IL-13 after clonal expansion and activation. Th2 cells cooperate with follicular T helper cells to induce immunoglobulin class switching in B cells under stimulation with IL-4 and IL-13, thereby enhancing the production of allergen-specific IgE and IgE memory B cells [32,33]. Allergen-specific IgE binds to the surface of effector cells (i.e., mast cells and basophils) via the high-affinity IgE receptor (FcεRI). Memory pools of allergen-specific Th2 cells and B cells are also generated during this phase (Figure 3).

Meanwhile, innate lymphoid cells (ILCs), particularly ILC2, are lymphocytes with helper activities that are involved in innate immunity. In contrast to Th cells, which require several days for differentiation, ILCs reside as mature lymphocytes in steady-state tissue and are rapidly and antigen-nonspecifically activated by EC-derived TSLP, IL-25, and IL-33 to induce type 2 immune responses, including IL-5 and IL-13 responses. ILC2 serves as the initiation point for the onset of allergic diseases, and has also recently been shown to induce Th2 cell differentiation. Thus, ILC2 is involved not only in the development of allergic diseases, but also in disease exacerbation and chronicity [34,35,36].

### 2.3. Immune Response of Effector Phases in Atopic March

Once sensitization to an allergen is established, re-exposure to the same allergen induces degranulation of inflammatory mediators upon crosslinking of FcεRI receptor-bound specific IgE on mast cells and basophils. This induces an immediate-phase reaction and acute inflammation, followed by a late-phase allergic reaction through activation of memory allergen-specific Th2 (T_MEM_) cells [35]. This process is accompanied by the infiltration or formation of tertiary lymphatic tissues in the dermis of AD, as well as in the submucosal tissues of AA.

The accumulation of effector cell-derived mediators and activation of T_MEM_ cells with antigen presentation by DCs and B cells results in an abundance of IL-4, IL-5, IL-9, IL-13, and IL-31 in lesions. This results in maintaining allergen-specific IgE levels, eosinophilia, and the recruitment of inflammatory cells to inflamed tissue, all of which induce tissue damage and increase mucus production and airway hyperresponsiveness in AA (Figure 4). This process also induces changes in the skin microbiome and reduces filaggrin expression, thereby accelerating impairment of skin barrier function in AD [33,36,37]. Furthermore, IL-31 affects sensory nerves and results in marked pruritus. Scratching behavior due to pruritus further worsens barrier damage. Then, TSLP, IL-25, and IL-33 are released from ECs, further accelerating Th2 immune skewing via ILC2 [29].

The T_MEM_ cells generated in the sensitization phase are allergen-specific central memory T (T_CM_) cells and effector memory T (T_EM_) cells, and their immunological memory is maintained for a long period. T_CM_ cells reside mainly in the T cell region of secondary lymphoid tissue. T_CM_ cells proliferate rapidly and differentiate partly into T_EM_ cells by re-exposure to the same allergen. T_EM_ cells reside mainly in local lesions (e.g., the lung and gut) and can produce high levels of Th2 cytokines when re-exposed to the allergen. These T_MEM_ cells make it possible to activate a rapid immune response. T_MEM_ cells circulate and infiltrate the skin, exacerbate AD, and enter the systemic circulation, where they spread to remote organs [2,29,38]. Then, upon re-exposure to allergens in individuals previously sensitized to the same allergens, diverse atopic disorders are initiated, resulting in the induction of atopic march.

## 3. Possible Involvement of Increased Oxidative Stress Status in the Atopic March

Oxidative stress is defined as an imbalance between the systemic manifestation of reactive oxygen species (ROS) and the ability of the biological system to readily detoxify reactive intermediates or to repair the resulting damage [37,38]. ROS are chemically reactive entities containing oxygen, either as oxyradicals or nonradical species, such as superoxide (O_2_^−^·), hydroxyl (OH·), alkoxyl (RO·), peroxyl (ROO·) radicals, hydrogen peroxide (H_2_O_2_), nitric oxide (NO·), and peroxynitrite (ONOO^−^). ROS are produced by all cell types, such as neutrophils, eosinophils, monocytes, macrophages, cytotoxic lymphocytes, epithelial cells, endothelial cells, and other resident cells. ROS can be formed by the action of many enzymes. Any excess of ROS is strictly limited by endogenous antioxidative defense mechanisms, which contain various enzymes, proteins, and low-molecular-weight molecules [39].

Disturbances in the normal redox state of cells can cause toxic effects and damage all cellular components, including lipids, proteins, and DNA. Some ROS act as cellular messengers in redox signaling [40,41]; therefore, oxidative stress can cause perturbations in redox signaling mechanisms that control various cellular functions, such as enzyme activation/inhibition, membrane signal transduction, transcription factor binding/gene expression, proliferation/apoptosis, and precursor cell ontogeny.

In the preceding sections, the concept of “atopic march” provides a perceptive for the mechanistic research, prediction, prevention, and treatment of allergic diseases [9,42]. The mechanisms involved in atopic march, from AD in infancy to other allergic diseases in later childhood such as AA and AR, likely arise from the induction of local and systemic type 2 immune responses via epicutaneous allergen sensitization. These findings support the view that AD is not merely a disease confined to the skin, but is a systemic disease. The inflammatory responses induced by AD are manifested by increased production of type 2 cytokines, such as IL-4, IL-13, IL-25, IL-33, and TSLP [43]. The major characteristics of these responses are taken over by AA and AR, which manifests the critical feature mediating the progression of systemic atopy [44].

The etiopathogenesis of atopic march is multifactorial; both the abovementioned dysregulated immune responses and genetic and environmental factors are implicated in its progression [9,42]. Because ROS promote tissue inflammation, barrier defects, and upregulation of genes encoding proinflammatory cytokines, oxidative stress likely contributes to the progression of atopic march [45,46,47]. The measurement of various markers for oxidative damage and antioxidants helps to clarify the role of oxidative stress in a variety of pediatric diseases [39,48]. In previous studies, oxidative stress biomarkers were determined in samples of blood, urine, nasal lavage fluid, bronchoalveolar lavage fluid, and exhaled breath condensate (EBC) in children with various allergic conditions [39,49,50]. A summary of these pointe is presented in Figure 5.

Herein, we refer to leading articles to investigate the role of oxidative stress in atopic march. For children with AD, a previous study indicated that patients with AD had significantly higher levels of serum malondialdehyde (MDA), a marker of lipid peroxidation, as compared with healthy controls, in addition to significantly lower levels of antioxidants such as superoxide dismutase, glutathione peroxidase, catalase, and vitamins A, C, and E [51]. The oxidative stress biomarkers of urinary 8-hydroxy-2′-deoxyguanosine (8-OHdG, a marker of oxidative DNA damage), acrolein-lysine (a marker of lipid peroxidation), and bilirubin oxidative metabolites (a marker of antioxidant bilirubin oxidation) were significantly higher in children with AD than in control subjects [52,53]. It was recently found that urinary 8-OHdG levels greater than the 75th percentile were associated with a higher risk of AA as compared with a reference group of the 0–25th percentiles [54]. In addition, it was found that the dynamic thiol-disulfide balance in the patient group was weakened, and the balance shifted towards the oxidative side as compared with a control group. The total oxidant status, disulfide (SS), and disulfide (SS)/total thiol (-SH plus -S-S-) ratios were significantly increased in blood samples from the patient group, while native thiol (-SH) and the native thiol (-SH)/total thiol (-SH plus -S-S-) ratio were significantly decreased [55]. Interestingly, Peroni et al. reported a significant decrease in pH and an increase in leukotriene B4 and 8-isoprostane (a marker of lipid peroxidation) in EBC in children with AD, suggesting the existence of airway inflammation in these patients [56].

The role of ROS and oxidative stress in the pathogenesis of AA has been studied for many years. The lungs are vulnerable to the effects of oxidative stress due to persistent exposure to allergens and environmental pollutants [46,57,58,59]. During exacerbation, asthmatic children had significantly higher serum MDA and significantly lower serum antioxidants, such as vitamin C, vitamin E, and uric acid as compared with controls. A significantly negative correlation between MDA and vitamin C was observed in severe asthmatic attacks [60]. Significantly lower glutathione peroxidase activity in erythrocytes was also found in children with controlled asthma as compared with healthy controls, suggesting an association between chronically increased oxidative stress and asthma in childhood [61]. Significantly increased levels of 8-isoprostane and proinflammatory leukotrienes were found in the blood, bronchoalveolar lavage fluid, and EBC of asthmatic children in several studies, and these increased levels tended to be directly correlated with asthma severity [62,63,64,65]. In addition, the MDA levels were significantly higher and reduced glutathione levels were significantly lower in both nasal and oral EBC samples of asthmatic children than those of healthy controls [61,66].

For AR, changes in the studied oxidative biomarkers also indicated increased oxidative stress status in children with AR. The levels of MDA in erythrocytes and hydroperoxides in plasma were significantly increased, and those of superoxide dismutase and catalase in erythrocytes and total antioxidant status in plasma were significantly decreased in the patients as compared with healthy controls [67]. Total oxidant status and total peroxide concentration in plasma were both significantly higher in children with AR than in healthy controls [68]. MDA levels were significantly higher and reduced glutathione levels were significantly lower than those in healthy controls in both nasal and EBC samples [66]. The plasma paraoxonase 1 levels were significantly lower in children with AR than in controls and the total oxidant status levels were significantly higher. Significantly negative and positive correlations were observed between these respective parameters and nasal symptom scores in the patients [69].

The above results in previous studies likely confirm the hypothesis that sustained oxidative stress contributes to atopic march. Considering that increased oxidative stress is part of the driving mechanism for chronic inflammation and disease progression and exacerbation, targeting oxidative stress with antioxidants might be a logical approach to attenuate or halt atopic march [58,70]. Reduction of oxidative burden on these children using a variety of nutritional and pharmacological approaches might hold promise for this purpose. However, before antioxidative therapy is accepted in clinical practice, it is necessary to characterize children at high risk for atopic march using specific oxidative biomarkers (e.g., blood or urinary levels of 8-OHdG or 8-isoprostane, blood levels of MDA, total hydroperoxides, dynamic thiol-disulfide balance, or antioxidative enzymes in plasma or erythrocytes). Furthermore, longitudinal studies should be conducted to evaluate panels of oxidative stress biomarkers together with traditional clinical endpoints in these pediatric patients [39].

## 4. Atopic March and Challenges in Epidemiological Studies

According to the observations of epidemiological studies, AD in early infancy is followed by AA and AR later in life; thus, atopic march was proposed to describe the typical order of progression of the development of AD, AA, and AR. For example, Sporik et al. followed a cohort of children at risk of atopic disease and reported a decrease in the prevalence of eczema and an increase in the prevalence of AA and AR with age [71]. Moreover, using a birth cohort study of 826 children, Martinez et al. reported that eczema in the first year of life was an independent risk factor for persistent wheezing [72]. Although AA and AR generally develop later in atopic march than AD, various epidemiological studies have also strongly suggested a correlation between AA and AR [73]. For example, Shaaban et al. proposed that AR is a strong predictor of AA in adults, even in the absence of AD [74]. Furthermore, a cohort study of children with AA alone suggested the existence of a “reverse” atopic march in which AA precedes AD [75]. Others have proposed that FA may also be related to AA and/or AR, either through AD or a different mechanism [26]. Thus, the interrelationships among pediatric allergic diseases are more complex than initially suggested in early epidemiological studies.

In recent years, longitudinal epidemiological studies, including larger birth cohorts, have shown that the sequence of progression of allergic disease at the individual level is complex and varied. According to a review of 13 prospective cohort studies, only one in three children who presented with eczema in infancy developed AA in early childhood [76]. This may be related to the complexity of the situation, wherein each allergic disease is composed of different phenotypes. To address these issues and determine a trajectory that predicts the prognosis of individual patients, it is necessary to specify the phenotype of each allergic disease. Mechanisms of the Development of Allergy (MeDALL), the Seventh Framework Program (FP7) of the European Union project, integrated several European birth cohorts and investigated the complexity of these allergy phenotypes using unsupervised statistical models, such as exploratory factor analysis, hierarchical clustering, and latent class analysis (LCA). MeDALL proposed a new vision of multimorbidity independent of IgE sensitization and showed that monosensitization and polysensitization are two distinct phenotypes [77]. Subsequently, latent class growth analysis (LCGA) and longitudinal latent class analysis (LLCA) identified 3–5 different wheeze trajectory groups among children aged 0–18 years [78].

Recently, Ödling et al. applied LCA to AA trajectories from infancy to young adulthood and identified four distinct trajectory groups and their background factors: class 1 (no/infrequent AA) had the highest proportion of breastfeeding for more than 4 months, class 2 (early onset transient AA) had the highest proportion of parental smoking, and class 3 (adolescent-onset AA) had the highest proportion of women. Meanwhile, class 4 (persistent AA) had the lowest proportion of women, and almost half had a family history of allergic disease. The study reported that participants with early allergic comorbidities fell into the class 3 or 4 trajectory groups, which is consistent with the phenomenon of atopic march. One-third of classes 3 and 4 had eczema in early childhood and were sensitive to food allergens. More than two-thirds of both classes were sensitized to inhalant allergens during adolescence and young adulthood [79]. By identifying and systematizing the different phenotypes of each allergic disease in this way, it is possible to predict the sequence of allergic disease progression more accurately at the individual level.

In addition to identifying the variations in the possible complex pathways, it is important to understand how genetics and environmental factors contribute to each pathway. Once the factors that can be used for intervention have been identified, steps can potentially be taken to prevent the progression of allergic diseases. The Multicenter Allergy Study (MAS) in Germany highlighted the role of risk from allergen exposure and identified a number of possible environmental factors, such as viral infections and indoor allergens. For example, children sensitized to airborne allergens before 3 years of age had a substantially increased risk of AA and lower lung function in later life [80]. Cluster analysis revealed that the risk of AA was highest in the group sensitized to airborne allergens by 1 year of age [81]. In addition, the Childhood Origins of ASThma (COAST) cohort and others showed that preschool wheezing associated with rhinovirus infection significantly increased the risk of future AA [82,83]. Family or twin studies are ideally suited to assessing the respective contributions of genetics and the environment. Kahn et al. conducted a systematic review of sibship and twin data and showed that genetics played a greater role than environment in the progression from eczema to AA and/or AR, and that the link between eczema and AA and AR was independent of shared environmental factors in early life [84].

The phenotypic diversity and interrelationships of allergic diseases make it challenging to elucidate their trajectories and predict individual prognoses (Figure 6). However, given the recent rapid increase in the prevalence of allergic diseases, systematizing the trajectory of allergic diseases in the light of phenotypes and research on genetic and environmental factors is a high priority. Recent epidemiological studies using big data have the potential to contribute to this area.

## 5. Prevention of Atopic March

### 5.1. Prevention of the Onset of AD

AD has been reported to increase the risk of developing other allergic diseases, that is, AD can be the starting point for allergic marches [5]. To date, clinical studies have been conducted to prevent the onset of AD, such as food removal during pregnancy and lactation [85], long-term breastfeeding [86], ingestion of hydrolyzed milk during infancy [87], and environmental improvement aimed at avoiding tick antigens [88]. However, their effectiveness has been denied by meta-analysis. Allergen immunotherapy for mites in high-risk babies also did not prevent the onset of AD [89].

The effectiveness of nutritional therapies such as vitamin D, fish oil, and probiotics has also been examined. A correlation between a decrease in blood vitamin D level and an increase in the incidence of AD was reported, but a randomized comparative study on the prevention of AD by vitamin D supplementation showed no preventive effect [90]. Oral administration of fish oil to high-risk pregnant women suppressed the onset of AD at 1 year of age, but there was no significant difference at 3 years of age. In addition, it has been reported that administration of probiotics reduced the risk of developing AD [91]. The World Allergy Organization Guidelines Panel suggests using probiotics for pregnant women and infants at high risk of allergies, but the evidence of benefit is still uncertain.

AD is triggered by barrier dysfunction and environmental factors. According to the barrier dysfunction due to mutation of the filaggrin gene and dual allergen exposure hypothesis, it has been noted that skin barrier protection is effective as a preventive method for AD. On the one hand, in a randomized controlled trial (RCT) of high-risk newborns with a family history of AD in Japan, the prevalence of AD at 32 weeks of age was suppressed by 32% in a group in which the moisturizer was applied to the whole body every day as compared with a control group [92]. In addition, in the United Kingdom (UK) and the United States RCTs of high-risk newborns within 3 weeks of age, the cumulative incidence of AD at 6 months of age was reduced in a group with systemic moisturizer application as compared to a control group [93]. In a comparative study of newborns, the use of moisturizers up to 3 months of age reduced infancy AD [94]. In the Barrier Enhancement for Eczema Prevention (BEEP) study, skin care by systemic use of moisturizers started from the neonatal period significantly suppressed the onset of AD at 6 months of age as compared with a non-use group [95]. On the other hand, in the Preventing Atopic Dermatitis and ALLergies in childhood (PreventADALL) study, systemic use of moisturizers in high-risk infants from the neonatal period could not prevent the onset of AD [96]. In a pilot study of the PEBBLES study in Australia [97], a protocol using a moisturizer containing ceramide twice a day suppressed the sensitization of food allergens at 12 months of age, although there was no significant difference [98]. Differences in these RCT results are thought to be due to differences in intervention methods for skin barrier disorders and differences in the background of the target child. More detailed RCTs are desired in the future.

### 5.2. Prevention of the Onset of FA

Some children that develop AD during infancy are often already sensitized to food antigens. It is also common for fully breastfed infants to be sensitized to eggs and milk before the start of weaning food. Previously, it was expected that mothers could prevent infant AD and FA by preventing sensitization to food antigens during the fetal and lactation periods. However, the removal of food from pregnant and lactating mothers did not prevent the sensitization of food antigens in the offspring or the development of AD [83,97,98]. It is not recommended for mothers to remove food during pregnancy or lactation to prevent the development of FA, because food removal can lead to harmful nutritional disorders for the mother and baby.

Many children with FA have AD in infancy, suggesting a decrease in skin barrier function and the presence of eczema as a risk factor for percutaneous sensitization to food antigens. Skin care for AD may suppress the development of allergen sensitization and FA. Unfortunately, in an RCT in Japan, there was no significant difference in egg white/ovomucoid sensitization rates at 32 weeks of age due to infancy skin care interventions [92].

The results of a birth cohort study clarified that delaying feeding babies baby food increases the incidence of FA [99]. A comparative study in Israel and the UK also showed that the UK, which started peanuts as a weaning diet later, had a 10-fold higher prevalence of peanut allergies than Israel, which started earlier [100]. In an Australian cohort study, the odds ratios were 4.4 for children who started eating eggs at 7 months of age and 5.9 for children who started eating eggs at 12 months of age, as compared with those who started eating eggs at 4 months of age [101]. A cohort study in Israel showed that infants who started milk containing milk protein after 2 weeks of age had a higher risk of milk allergies than those who started milk containing milk protein by 2 weeks of age [102]. It has also been reported that infants who were completely breastfed until the first month of life had significantly more milk allergies, and infants who had been fed artificial milk daily from an early stage had significantly fewer milk allergies [103]. Delaying the onset of certain foods for high-risk babies is not currently recommended, because it does not reduce the risk of developing the disease.

On the basis of the findings of observational studies that delaying baby food increases FA, various RCTs have been conducted on the effect of early food antigen intake on the prevention of FA. In the Enquiring About Tolerance study in the UK, the rates of sensitization and onset of FA were compared between a group of six types of foods (milk, chicken eggs, wheat, sesame, fish, and peanuts) for children who were completely breast-fed, a group that started ingestion by 5 months of age, and a group who started ingestion after 6 months of age. As a result of analysis only in infants who could comply with the protocol, a decrease in the sensitization and incidence rates of chicken eggs and peanuts were observed in the early introduction group [104]. However, more than 60% of the participants dropped out, and no clear conclusion was reached regarding the early start of baby food. However, in a UK LEAP study of high-risk infants, the prevalence of peanut allergies at 5 years of age was lower in the early peanut intake group than in the non-ingestion group (17.2% vs. 3.2%) [105,106]. Currently, in countries where the risk of developing peanut allergies is high, peanut intake should be started as soon as possible during the weaning period of babies [107].

Furthermore, for chicken egg allergy, several RCTs have been conducted to verify the prevention of onset by early ingestion. In the Australian STAR study, infants with a history of AD who received raw egg powder from 4 months of age had a lower incidence of egg allergy at 12 months of age than those who did not. However, this result was not statistically significant, and 30% of the infants developed allergic symptoms due to raw egg intake [108]. In addition, the Australian STEP study investigated the prevalence of egg allergies at 12 months of age between an intake group and a non-ingestion group of raw egg powder in infants aged 4–6 months who had no history of allergies; however, there was no significant difference [109]. In a HEAP study in Germany, there was no significant difference in the incidence of chicken egg allergy between a group receiving a moderate amount of raw eggs and a placebo group in infants who were not sensitized to eggs [110]. In an Australian BEAT study in 4-month-old infants who were not sensitized to eggs, there was no significant difference in the incidence of chicken egg allergy between a group receiving a small amount (350 mg) of raw eggs and a placebo group [111].

However, a PETIT study in Japan conducted an RCT of 4-month-old babies with AD, who maintained eczema remission with aggressive skin care [112]. The incidence of egg allergy was significantly reduced in a group that started ingesting a small amount of dried egg powder from 6 months of age as compared with a group in which the intake of eggs was removed until 12 months of age (8.3% vs. 37.7%). This study was conducted safely, without any allergic reactions due to egg intake. It is speculated that the combined use of aggressive skin care has a preventive effect on the onset of egg allergy. A systematic review that comprehensively analyzed multiple RCTs concluded that early weaning (4–6 months) intake of eggs and peanuts reduced the risk of developing egg and peanut allergies. The American Academy of Pediatrics found that delaying the introduction of allergenic foods beyond the introduction of baby food at 4–6 months of age could prevent atopic disease [113]. In particular, for peanuts, early intake can prevent allergies. In addition, the European Society of Pediatric Gastroenterological and Liver Nutrition states that baby food should be started 4–6 months after birth, and foods with high allergenicity may be introduced after 4 months [114]. Currently, weaning guidelines in various countries and regions have similar descriptions (Table 1).

## 6. Conclusions

Because there are complex interrelationships among pediatric allergic diseases in atopic march, it is important to not only understand the genetic and environmental factors involved in each allergic disease, but also to elucidate the causes of allergic diseases from the perspective of atopic march as a whole. In this article, we have outlined atopic march from various perspectives including not only skin barrier dysfunction and immunological pathology, but also oxidative stress involvement, epidemiological classification of each phenotype, and effective early intake of causative foods. In the future, epidemiological studies using big data and various research approaches to investigate factors involved in the onset and progression of atopic march will lead to new preventative measures for this phenomenon.

## Figures and Tables

**Figure 1 children-08-01067-f001:**
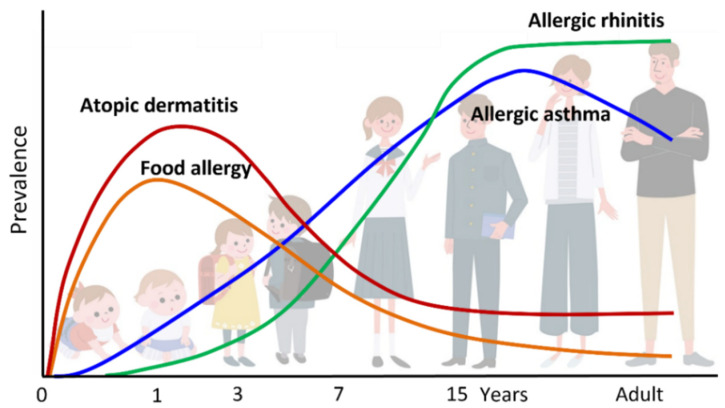
Atopic march. Atopic dermatitis (AD) generally develops first, followed by IgE-mediated food allergy (FA), allergic asthma (AA), and allergic rhinitis (AR). Development of FA, AA, and AR correlates with AD severity in infancy.

**Figure 2 children-08-01067-f002:**
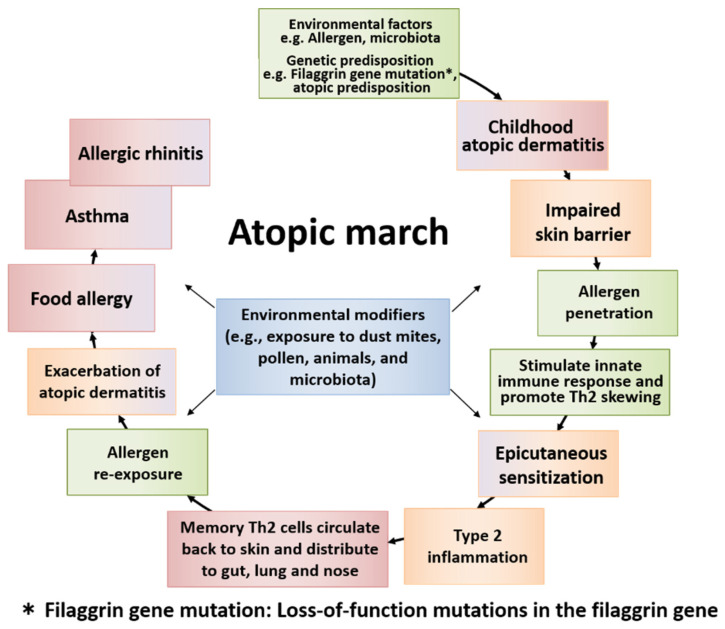
The general course of atopic march and the temporal sequence of major factors involved. Atopic march is the clinical history of atopic diseases in different organs, wherein different causal allergens develop sequentially with age. Some symptoms become more prominent over time, while others subside. Atopic dermatitis (AD) generally develops first based on genetic predisposition and/or environmental factors. The skin barrier dysfunction derived from AD easily allows the allergen to penetrate the body, causing epicutaneous sensitization and type 2 inflammation in some children. Then, memory T helper 2 (Th2) cells circulate back to the skin and exacerbate AD. The cells then distribute to the gut, lung, and nose. Patients develop food allergy, allergic asthma, and allergic rhinitis with increased sensitization to food and/or environmental allergens, resulting in the induction of atopic march.

**Figure 3 children-08-01067-f003:**
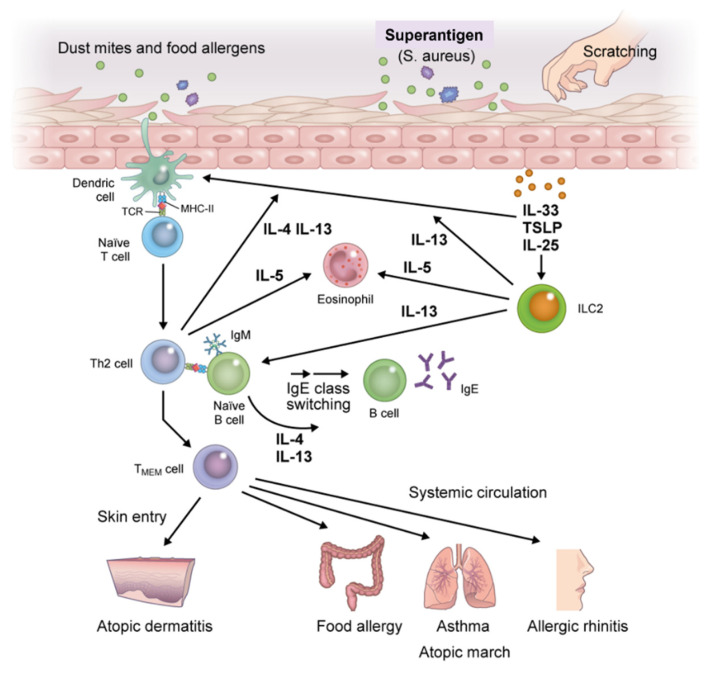
Immunologic mechanisms underlying atopic march. Exposure of skin with impaired barrier function to food allergens, dust mites, or mechanical damage leads to the release of epithelial cell-derived thymic stromal lymphopoietin (TSLP), interleukin (IL)-25, and IL-33, which induce the activation of immature dendritic cells (DCs) and group 2 innate lymphoid cells (ILC2s). DCs that capture allergens migrate to draining lymph nodes, process allergens, and present them to naïve T cells which promote the generation of allergen-specific Th2 cells. Th2 cells produce high levels of IL-4 and IL-13 after clonal differentiation and expansion and induce B cell isotype switching to specific IgE cells, thereby enhancing the production of allergen-specific IgE and IgE memory B cells. Allergen-specific IgE binds to the surface of effector cells (i.e., mast cells and basophils) via the high-affinity IgE receptor (FcεRI). ILC2 activation further accelerates antigen-nonspecific Th2 immune skewing; memory pools of allergen-specific Th2 and B cells are also generated during this phase. Memory allergen-specific Th2 cells circulate and infiltrate the skin, provoking exacerbation of AD, and entering the systemic circulation where they spread to remote organs. Then, upon re-exposure to allergens in individuals previously sensitized to the same allergens, diverse atopic disorders are initiated, resulting in atopic march. Abbreviation: T_MEM_, memory allergen-specific T helper 2.

**Figure 4 children-08-01067-f004:**
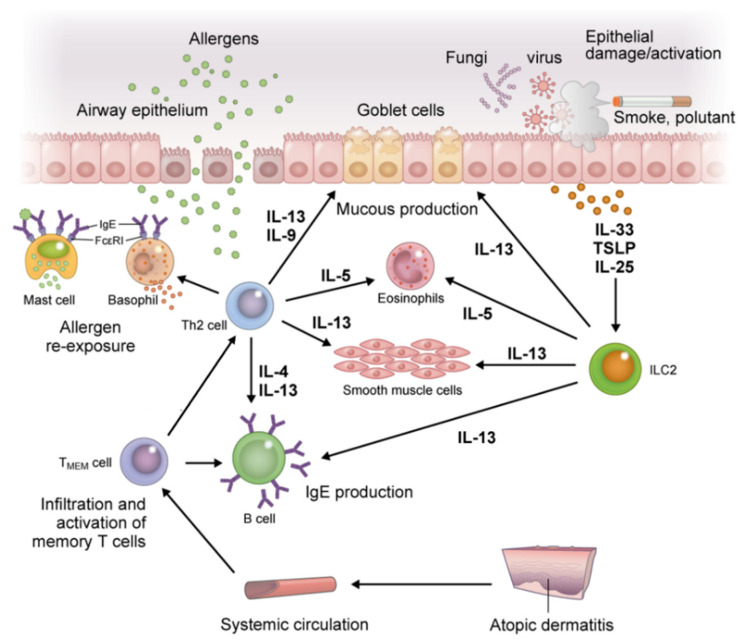
Inflammation in the respiratory tract, a distant organ from the skin in atopic march. Re-exposure to a previously sensitized allergen induces degranulation of inflammatory mediators upon crosslinking of FcεRI receptor-bound specific IgE on mast cells and basophils. This induces an immediate-phase reaction and acute inflammation, followed by a late-phase allergic reaction through activation of T_MEM_ cells. The T_MEM_ cells produce IL-4, IL-5, IL-9, and IL-13 in cooperation with ILC2s, and lead to maintenance of allergen-specific IgE levels, eosinophilia, and recruitment of inflammatory cells to inflamed tissue, all of which induce tissue damage and increase mucus production and airway hyperresponsiveness in allergic asthma. Abbreviations: IL, interleukin; ILC2s, group 2 innate lymphoid cells; T_MEM_, memory allergen-specific T helper 2.

**Figure 5 children-08-01067-f005:**
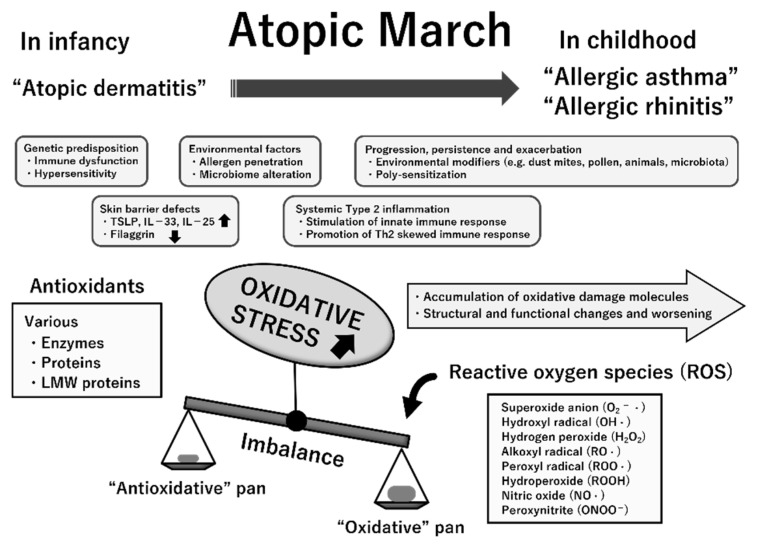
Multifactorial etiopathogenesis and possible involvement of increased oxidative stress status in atopic march. The temporal pattern of atopic march is generally from atopic dermatitis in infancy to gradual development into allergic asthma and allergic rhinitis in childhood. While its pathogenesis is complex, several essential mechanisms likely underlie atopic march. Oxidative stress is an imbalance between reactive oxygen species (ROS) generation and antioxidative defense mechanisms with any excess of the former, resulting in macromolecular damage and dysfunction. Chronically increased oxidative stress may contribute to the progression, persistence, and exacerbation of allergic inflammation, thereby resulting in atopic march. This suggestion is mostly based on previous clinical studies that have indicated the linkage of oxidative damage attributable to ROS (quantified by measurement of specific biomarkers) to the pathogenesis and progression of these atopic diseases. Abbreviations: TSLP, thymic stromal lymphopoietin; IL, interleukin; LMW, low-molecular-weight.

**Figure 6 children-08-01067-f006:**
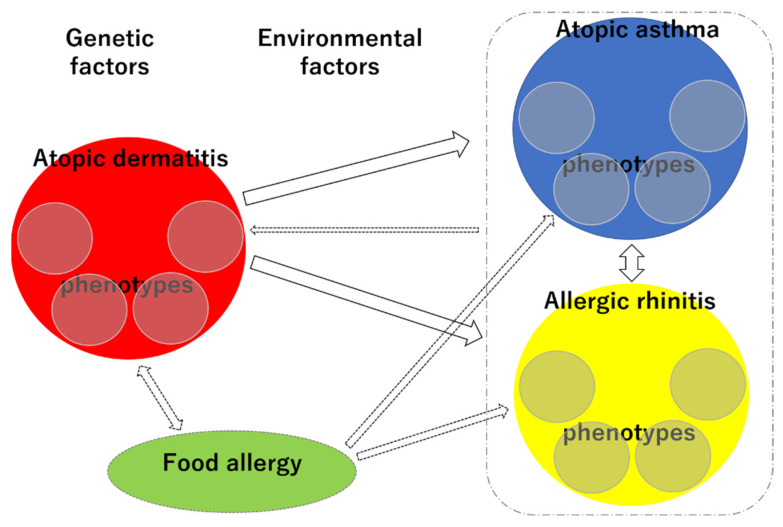
Schematic representation of the complex phenotypes and trajectories of each allergic disease. Small circles within red circle, each phenotype of AD; small circles within blue circle, each phenotype of AA; small circles within yellow circle, each phenotype of AR.

**Table 1 children-08-01067-t001:** Guidelines and recommendations for early intake of eggs and peanuts in each country.

Country	Organization	Targeted Children	Recommended Age to Start Peanuts	Recommended Age to Start Chicken Eggs	Reference
United States	NIAID	Eczema or egg allergy	4–6 months old	n.d.	[115]
European countries	ESPGHAN	Eczema or egg allergy	4–11 months old	n.d.	[114]
Australia	ASCIA	All children	By 12 months old	By 8–12 months old	[116]
Japan	JSPACI	Atopic dermatitis	n.d.	6 months old	[117]
Asian countries	APAPARI	Severe eczema	n.d.	4–6 months old	[118]
Canada	CPS	Eczema or family history of allergic diseases	6 months old	6 months old	[119]

Abbreviations: NIAID, National Institute of Allergy and Infectious Diseases; ESPGHAN, European Society for Paediatric Gastroenterology Hepatology and Nutrition; ASCIA, Australasian Society of Clinical Immunology and Allergy; JSPACI, Japanese Society of Pediatric Allergy and Clinical Immunology; CPS, Canadian Pediatric Society; n.d: not described.

## Data Availability

Not applicable.

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
