# Peer review of "Current Insights into Atopic March"

_children, 2021, doi:10.3390/children8111067_

Round 1

Reviewer 1 Report

Comments:

The review article “Current insights into atopic march” [children-1465954] provides a comprehensive understanding of “Atopic march” process that involves multiple immunological pathways. Atopic dermatitis generally develops first, followed by IgE-mediated food allergy, allergic asthma and allergic rhinitis.  The article is presented in a systematic way, interesting and is well written. Few minor suggestions could help to improve the quality of the manuscript:

Comment 1: Section 2.1: add a few line related to “Filaggrin” protein as filaggrin plays a critical role in the skin's barrier function.

Comment 2: It is important to include 3-4 more lines in the “conclusion” section that how your review is unique and adds extra knowledge in the scientific field.

Comment 3: Mentioned Figure 6 at the appropriate position in the text.

Author Response

Response to Reviewer 1 Comments

We are grateful to Reviewer 1 for the critical comments and useful suggestions that have helped us improve our paper. As indicated in the responses that follow, we have considered all these comments and suggestions when revising our manuscript.

Point 1: Section 2.1: add a few line related to “Filaggrin” protein as filaggrin plays a critical role in the skin's barrier function.

Response 1: Thank you so much for your suggestion. We added the description of the skin barrier function of filaggrin (Line 120-126).

Point 2: It is important to include 3-4 more lines in the “conclusion” section that how your review is unique and adds extra knowledge in the scientific field.

Response 2: Thank you very much for your kind and important suggestion. We added the description about uniqueness and scientific novelty of our review in the conclusion (Line 521-524).

Point 3: Mentioned Figure 6 at the appropriate position in the text.

Response 3: Thank you for your suggestion. We added the “Figure 6” to the appropriate position in the text (Line 387).

Reviewer 2 Report

REVIEW:

Current insights into atopic march

Overall an excellent review.

However they omit any reference to allergy and otitis media or eustachian tube dysfunction which has been shown to have a high correlation to allergic rhinitis.[1] Per the 2016 guidelines, the middle ear is part of the unified airway, and “like other parts of respiratory mucosa, the mucosa lining the middle ear cleft is capable of an allergic response”.[2] This might be included in their section 1.3

            Instituting Allergic immunotherapy (AIT) not only treats allergic diseases of the unified airway, but can actually prevent treated patients from developing asthma.[3]

            De Corso provides an in depth explanation of physiopathology factors linking allergy to increased risk of middle ear inflammation.[4] His recent systematic review of 3,010 papers found that “clinical evidence and analyses of biomarkers suggested that allergy may be linked to some phenotypes of otitis media and, in particular, to otitis media with effusion and acute re-exacerbations in children with middle ear effusion”.[4]

            Their section 2 is excellent.

They might have considered including a review of the cells involved in allergic disease. Atopy involves a type I IgE-mediated hypersensitivity reaction in which activated mast cells and eosinophils participate in a Th-2 driven inflammatory reaction.[5] Atopy has been defned as ``the propensity of an individual to develop IgE antibodies'', and this condition is to be distinguished from that of an atopic patient with related symptoms who is deemed ``allergic''.[6] Indicators of a Th2 driven allergic response, mast cells[7] with their mediator tryptase and degranulating eosinophils[8] are present in a majority of ears with chronic effusion – as well as in the sinuses and lungs of allergic individuals.

            The majority of atopic OME patients, unlike the nonatopics, demonstrated signifcant involvement of eosinophils and mast cells, as refected by increased

levels of both ECP and tryptase in their middle ear (P<0.001). ECP was elevated (>10 mg/l) among 68/79 (86.1%) atopics. Elevated ECP correlated to a patient's being atopic (Fisher, P=0.001) and was found to serve as a marker of atopy among

patients with nonpurulent effusion with a positive predictive value of 97.1% and a diagnostic sensitivity of 86% . Typtase was >2 mg/l in 23/36 (64%) among atopic patients.[8] Cell kinetics would suggest that these elevated levels of ECP and tryptase react in an active inflammatory process.[9]

            One more recent review of risk factors for Otitis Media from a total of 2971 articles were searched, and 198 full-text articles were assessed for eligibility; 24 studies were eligible for this meta-analysis. Regarding risk factors for COM/ROM, there were two to nine different studies from which the odds ratios (ORs) could be pooled. The presence of allergy or atopy increased the risk of COM/ROM (OR, 1.36; 95% CI, 1.13–1.64; P = 0.001).[10]

            My suggestion would only be to additionally mention otitis media as an additional part of the allergic march and perhaps include some of these references.

Or, this might make for an additional paper to supplement this otherwise excellent review.

  1. Roditi RE, Veling M, Shin JJ. Age: An effect modifier of the association between allergic rhinitis and Otitis media with effusion. Laryngoscope 2016; 126:1687-1692.
  2. Rosenfeld RM, Shin JJ, Schwartz SRet al. Clinical Practice Guideline: Otitis Media with Effusion Executive Summary (Update). Otolaryngol Head Neck Surg 2016; 154:201-214.
  3. Jacobsen L, Niggemann B, Dreborg Set al. Specific immunotherapy has long-term preventive effect of seasonal and perennial asthma: 10-year follow-up on the PAT study. Allergy 2007; 62:943-948.
  4. De Corso E, Catone E, Galli J; et al. Otitis media in children: which phenotypes are most linked to allergy? A systematic review. Authorea 2020.
  5. Pauwels R. Future of anti-inflammatory therapy in asthma. Allergy 1995; 50 (suppl 22):27-31.
  6. Pepys J. "Atopy": a study in definition. Allergy 1994; 49:397-399.
  7. Hurst DS, Amin K, Sevéus L, Venge P. Evidence of mast cell activity in the middle ear of children with otitis media with effusion. Laryngoscope 1999; 109:471-477.
  8. Hurst DS, Venge P. Evidence of eosinophil, neutrophil, and mast-cell mediators in the effusion of OME patients with and without atopy. Allergy 2000; 55:435-441.
  9. Demoly P, Crampette L, Mondain Met al. Assessment of inflammation in noninfectious chronic maxillary sinusitis. J Allergy Clin Immunol 1994; 94:95-108.
  10. Zhang Y, Xu M, Zhang J, Zeng L, Wang Y, Zheng QY. Risk factors for chronic and recurrent otitis media-a meta-analysis. PLoS One. 2014;9(1):e86397. Published 2014 Jan 23. doi:10.1371/journal.pone.0086397

Author Response

Response to Reviewer 2 Comments

We appreciate Reviewer 2 for taking the time to offer us the critical comments and useful suggestions that have helped us improve our paper. As indicated in the responses that follow, we have considered all these comments and suggestions when revising our manuscript.

Point 1: My suggestion would only be to additionally mention otitis media as an additional part of the allergic march and perhaps include some of these references.

Response 1: Thank you for your suggestion. We added the description about otitis media as an additional part of atopic march, and added the related references to this sentences (Line 97-101).